# Mechanically tunable conductive interpenetrating network hydrogels that mimic the elastic moduli of biological tissue

Vivian R. Feig [1], Helen Tran[2], Minah Lee [2] & Zhenan Bao [2]

Conductive and stretchable materials that match the elastic moduli of biological tissue (0.5–500 kPa) are desired for enhanced interfacial and mechanical stability. Compared with inorganic and dry polymeric conductors, hydrogels made with conducting polymers are promising soft electrode materials due to their high water content. Nevertheless, most conducting polymer-based hydrogels sacrifice electronic performance to obtain useful mechanical properties. Here we report a method that overcomes this limitation using two interpenetrating hydrogel networks, one of which is formed by the gelation of the conducting polymer PEDOT:PSS. Due to the connectivity of the PEDOT:PSS network, conductivities up to 23 S m$^{-1}$ are achieved, a record for stretchable PEDOT:PSS-based hydrogels. Meanwhile, the low concentration of PEDOT:PSS enables orthogonal control over the composite mechanical properties using a secondary polymer network. We demonstrate tunability of the elastic modulus over three biologically relevant orders of magnitude without compromising stretchability ( > 100%) or conductivity ( > 10 S m$^{-1}$).

[1] Department of Material Science and Engineering, Stanford University, 443 Via Ortega, Room 307, Stanford, CA 94305, USA. [2] Department of Chemical Engineering, Stanford University, 443 Via Ortega, Room 307, Stanford, CA 94305, USA. Correspondence and requests for materials should be addressed to Z.B. (email: zbao@stanford.edu)

Electronics with tissue-like properties enable a natural integration between electronic functionality and the biological world. Such integration is increasingly important for treating numerous medical diseases like Parkinson's, for which electrical deep-brain stimulation has proven to be highly effective[1]. Moreover, electronic interfaces with biological tissue provide valuable scientific and diagnostic insights into complex medical phenomena, ranging from atrial fibrillation[2] to Alzheimer's[3,4]. However, conventional conductive materials have difficulty forming long-term stable and conformal interfaces with many biological tissues due to severe mechanical mismatch between the two material types. For example, brain tissue typically has elastic moduli < 1 kPa[5], whereas common electrode materials like silicon and tungsten have moduli of 50 GPa and 130 GPa, respectively[6]. Even conducting polymers (CPs), which have superior flexibility compared to inorganic conductors, typically have moduli in the 1 GPa range[6]. Furthermore, interfacing materials should be stretchable enough to maintain conformal contact with dynamic tissue surfaces. For example, the surface of the brain has an intrinsic displacement of over 10 μm from respiration alone[6]. Mechanical mismatch of tissue-interfacing materials can lead to reduced efficacy of both recording and stimulation, which may be further exacerbated by immunological responses that lead to scarring[6–8]. Thus, there is a strong demand for materials that can be rationally designed to possess mechanical properties that mimic tissue, without compromising their electronic performance. Ideally, such materials should be able to be tuned to form mechanically compliant interfaces with a wide range of biological tissues, from ultra-soft tissues such as the brain (0.5–1 kPa)[5] to stiffer tissues such as the skin[9] and certain regions of the heart (100–500 kPa)[10].

To mimic the mechanical properties of these tissues, hydrogels are a promising class of synthetic materials due to their high water content (70–99 wt%) that is similar to tissue[11]. Electrically conducting hydrogels made with CPs, including poly(3,4-ethylenedioxythiophene) polystyrene sulfonate (PEDOT:PSS)[12–15], polyaniline (PANI)[16–18], and polypyrrole (PPy)[19–22] have been reported, although there are few examples of mechanically robust gels that can stretch to conform to dynamic tissue surfaces. Among these CPs, PEDOT:PSS is advantageous because of its biocompatibility[23,24], and because it is commercially available in its doped form with reproducibly high conductivity, thus avoiding problems with batch to batch variation from in situ polymerization[25] and the use of alternative dopants that are less effective than PSS[26]. Reported examples of stretchable conductive hydrogels based on PEDOT:PSS either rely on in situ polymerization of EDOT within an inert hydrogel matrix[13,14,27], or involve blending PEDOT:PSS with hydrogel-forming precursors[15]. However, these strategies suffer from low conductivity (0.01–2.2 S m$^{-1}$) with relatively high PEDOT:PSS content up to 30 wt%, which makes it difficult to selectively tune the mechanical properties of the gel[13–15].

We hypothesize that the low performance of these gels arises from limited control of the CP network morphology, resulting in largely disconnected aggregates that require high loadings of CP to meet the percolation threshold for electronic conduction. Accordingly, we hypothesize that using gelation to ensure CP network connectivity should result in improved conductivity. Recently, it has been reported that PEDOT:PSS can form gels directly from aqueous solution, either by increasing the ionic strength[28], increasing concentration[28], or lowering pH[12]. Although these gels behave like solids, on their own they are highly brittle and difficult to handle. Still, the ability to form gels directly from commercially available PEDOT:PSS solutions presents a unique opportunity to control CP network connectivity to improve electronic conductivity.

In this work, we demonstrate a method for making conducting interpenetrating networks (C-IPNs) by infiltrating a loosely-crosslinked PEDOT:PSS gel with the precursors for a secondary polymer network (Fig. 1a). The controlled gelation of commercially available PEDOT:PSS enables us to achieve record-high conductivities up to 23 S m$^{-1}$, while the fact that PEDOT:PSS can gel at a low solids content of 1.1 wt% enables us to orthogonally control the mechanical properties of the C-IPN by tuning the secondary network properties. Using this method, we demonstrate that we can fabricate gels with ultra-soft moduli over three biologically relevant orders of magnitude (8–374 kPa) without compromising conductivity (> 10 S m$^{-1}$) or stretchability (> 100%).

## Results

**Synthesis and characterization of PEDOT:PSS hydrogels**. To synthesize C-IPN, PEDOT:PSS hydrogels were first formed from commercial aqueous dispersions of PEDOT:PSS with a low polymer content of approximately 1.1 wt% (11 mg mL$^{-1}$). Recently, Leaf and Muthukumar[28] proposed that aqueous PEDOT:PSS solution consists of large microgel particles which internally resemble a semi-dilute polyelectrolyte mesh and which have a low overlap concentration of 1 mg mL$^{-1}$. The authors found that increasing ionic strength can cause PEDOT:PSS solution to phase change into a solid-like gel, resulting in dramatically different rheological properties without changing the internal scattering profile of the microgels. As the primary interaction between microgel particles was found to be electrostatic in nature, the authors concluded that increased ionic strength helps screen the electrostatic repulsions between particles and thus allows them to form physical crosslinks between each other through π–π stacking of PEDOT. When a sufficient number of physical crosslinks form between particles, a macroscopically connected gel results[28]. Based on these findings, we hypothesized that the macroscopically connected microgel particles would lead to an increase in the number of connected pathways available for charge conduction, and should thus be a way to form highly conductive PEDOT:PSS hydrogels at a relatively low concentration.

To form PEDOT:PSS gels, we increased ionic strength of the solution using the ionic liquid 4-(3-butyl-1-imidazolio)-1-butanesulfonic acid triflate. A previous publication from our group found that this ionic liquid is an excellent dopant for PEDOT:PSS thin films, and that it interacts strongly enough with PEDOT:PSS to induce a significant change in the thin film morphology[29]. Based on these observations, we surmised that the same ionic liquid should also sufficiently interact electrostatically with PEDOT:PSS to push the solution past its gel phase boundary. Indeed, mixing the ionic liquid with PEDOT:PSS resulted in a solid-like gel, as indicated by the fact that the gel's storage modulus (G') exceed its loss modulus (G") in the frequency range of 0.1–100 Hz (Fig. 2a). Furthermore, the gel stiffness, given by G', increased with ionic strength (Fig. 2b), which is consistent with the proposed mechanism wherein electrostatic screening induces physical crosslinking. Within this rubbery regime, G' increased slightly with frequency, reflecting the dynamic nature of the physical crosslinks present in the gel (Fig. 2b). These rheological properties are consistent with those observed by Leaf and Muthukumar[28], who used both small salts and larger ionic liquids such as 1-ethyl-3-methylimidazolium tetracyanoborate (EMIM:TCB) to induce PEDOT:PSS gelation.

Interestingly, we found that the gelation rate, which is an important consideration for processability, could be tuned by the selection of the molecular additive. To investigate this, we mixed PEDOT:PSS with both our ionic liquid and a smaller metal salt

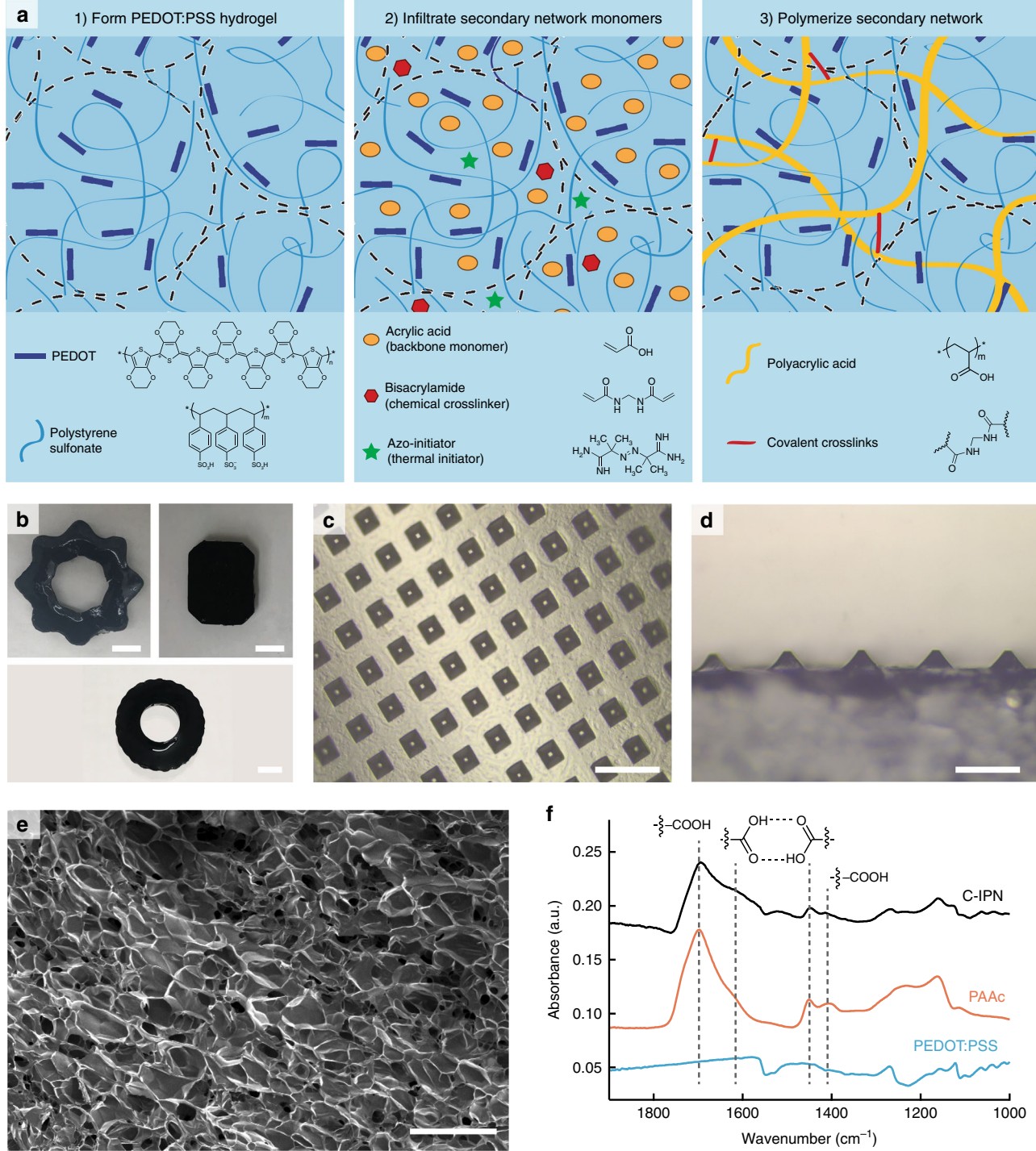

**Fig. 1** Fabrication and structure of C-IPN hydrogels. **a** Process for fabricating C-IPN hydrogels. First, PEDOT:PSS hydrogels are formed from aqueous solutions of PEDOT:PSS by using ionic liquid to screen the electrostatic repulsions between PEDOT:PSS microgels, enabling them to aggregate into a macroscopically connected network. Second, the PEDOT:PSS hydrogel is infiltrated with acrylic acid, bisacrylamide, and an azo-initiator. Finally, the polyacrylic acid network is formed by polymerizing the monomers in water at 70 °C. **b** Various large shapes made by casting the PEDOT:PSS/acrylic acid mixture into different silicone soap molds. Scale bars are 1 cm. **c**, **d** C-IPN can be micropatterned into pyramidal structures with features as small as 10 μm by casting into silicon molds. Scale bar is 200 μm **c** and 100 μm **d**. **e** Cross-sectional SEM image of freeze-dried C-IPN showing that the final gel is homogeneous and porous. Scale bar is 100 μm. **f** FTIR spectra of PEDOT:PSS, PAAc, and C-IPN, showing a clear presence of PAAc within the C-IPN composite

(CuCl₂) at the same overall ionic strength, and loaded the mixture onto a parallel plate rheometer oscillating at a constant strain rate of 1% and a frequency of 1 Hz. With the ionic liquid mixture, the crossover point at which *G' > G"* occurs after 15 min, before which the mixture can flow readily. By comparison, mixing PEDOT:PSS with CuCl₂ at an equivalent ionic strength resulted in nearly immediate gelation: by the time the solution was loaded onto the rheometer, *G'* already exceeded *G"* (Fig. 2c). The

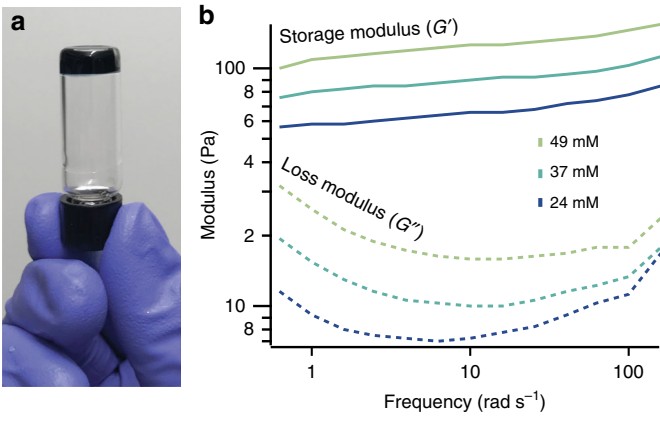

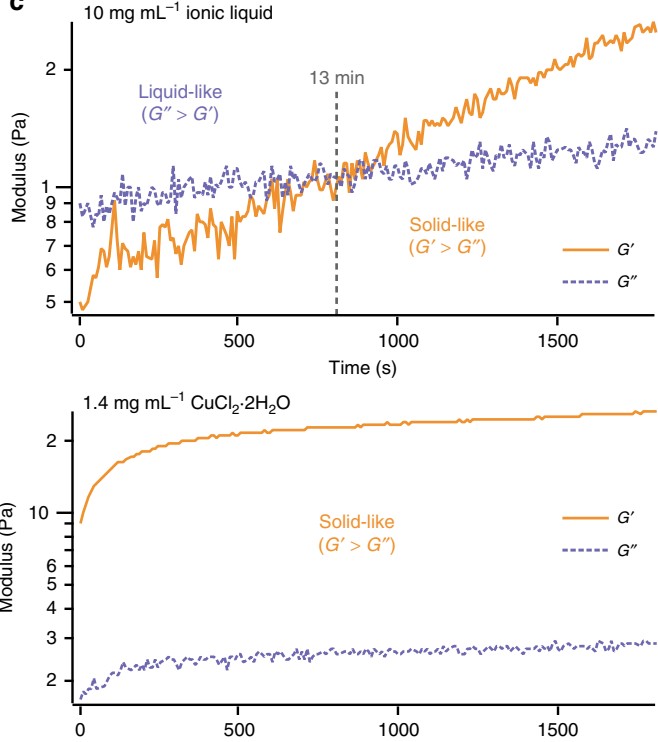

**Fig. 2** Gelation of PEDOT:PSS. **a**, **b** PEDOT:PSS forms a gel **a** after mixing it with ionic liquid. The gel strength, given by storage modulus ($G'$), increases with increasing ionic strength **b**. **c** Mixing PEDOT:PSS with either ionic liquid or CuCl$_2$ will cause the solution to gel, although the gelation occurs much more rapidly with CuCl$_2$. With ionic liquid, the PEDOT:PSS mixture only starts to become more solid-like ($G' > G''$) after 13 min. By contrast, the mixture with CuCl$_2$ gels so quickly that $G'$ already exceeds $G''$ by the start of the rheology measurement

widened window of liquid-like properties for the PEDOT:PSS/ionic liquid mixture enables us to more easily process it into different shapes (Fig. 1b). We could even process it into pyramidal structures with 10 μm resolution by casting the mixture into pre-fabricated silicon molds (Fig. 1c, d). To our knowledge, this is the first discussion of the kinetics surrounding the gelation behavior of PEDOT:PSS solutions.

**Synthesis and characterization of C-IPN gels.** Although the PEDOT:PSS hydrogels behave similar to solids, they are also

highly brittle. To improve mechanical properties, we selected polyacrylic acid (PAAc) as a secondary polymer network due to its biocompatibility[30,31] and high concentration of hydrogen bonding, which we hypothesized could further mechanically reinforce the C-IPN. In addition, aqueous PEDOT:PSS solution has been shown to gel when exposed to strong acids, and we surmised that infiltrating the PEDOT:PSS hydrogel with acidic monomers would result in additional reinforcement of the PEDOT:PSS matrix[12]. Finally, PAAc hydrogels can be easily tuned by varying the concentration of monomers in water and the ratio of bisacrylamide to acrylic acid, which controls the degree of covalent crosslinking (Supplementary Figure 1).

To infiltrate the PEDOT:PSS gels with PAAc monomers, the gels were soaked in an aqueous solution of acrylic acid, bisacrylamide, and a thermal radical polymerization initiator that activates above 60 °C. As expected, the acidity of the acrylic acid caused the PEDOT:PSS gels to shrink slightly during the exchange process (Supplementary Figure 2b), which qualitatively strengthened the PEDOT:PSS/acrylic acid composite even before the acrylic acid was polymerized. Finally, the acrylic acid network was polymerized within the PEDOT:PSS gel by placing the gel in a sealed container in an oven at 70 °C for 30 min. The thermal initiator was selected instead of a UV initiator due to its relatively fast kinetics and insensitivity to oxygen, and because it allows acrylic acid to be polymerized within the bulk PEDOT:PSS gel, which is opaque due to its electronically conducting nature.

To confirm the presence of the secondary network inside the C-IPN, Fourier-transform infrared spectroscopy was performed on C-IPN and compared with PEDOT:PSS and PAAc hydrogels (Fig. 1f). Peaks attributed to carboxylic acid and hydrogen bonding between carboxylic acid groups can be identified in both PAAc and C-IPN samples, indicating that the PAAc successfully was incorporated. In addition, scanning electron microscopic (SEM) imaging on the cross-section of a lyophilized C-IPN sample confirms that the gel is homogeneous and porous (Fig. 1e). Finally, X-ray photoelectron spectroscopy (XPS) measurements on PEDOT:PSS and C-IPN suggest that the PEDOT:PSS network is chemically unaffected by the presence of PAAc. In XPS spectra of PEDOT:PSS, two S2p peaks correspond to the presence of sulphur in PEDOT and in PSS, respectively. The relative size of these two peaks is proportional to the ratio of PEDOT to PSS, whereas the locations of the peaks correspond to the binding energy of the sulphur in its two respective chemical environments[29,32]. No significant change in peak area or position was observed in the PEDOT:PSS after PAAc was introduced (Supplementary Figure 2a).

Although ionic liquid is used to induce PEDOT:PSS gelation, we wanted to remove it from the final C-IPN gel because of its cytotoxicity. To facilitate diffusion of ionic liquid out of the gel, we soaked the PEDOT:PSS gels in deionized (DI) water and exchanged the water multiple times before infiltrating the PEDOT:PSS with acrylic acid. For our system, fluorine is a unique marker for the anionic component of the ionic liquid, yet no fluorine could be detected by XPS on the C-IPN. To confirm that the cationic component is also removed, we also performed $^1$H-nuclear magnetic resonance (NMR) spectroscopy on the solution obtained from washing the PEDOT:PSS gels with DI water and compared it with a spectrum of the neat ionic liquid (Supplementary Figure 3). The cation is clearly present in the wash solution. The broad peaks around 6.5–7.5 p.p.m. correspond to aromatic rings which are unique to PSS, suggesting that some PSS is also removed from washing the gel. The slight shifts in the peak at 6.2 p.p.m., which is attributed to the cation hydroxyl, may be due to the presence of PSS, since the rate of proton exchange between the hydroxyl group and solvent is highly dependent on acidic environment. Taken together, the XPS and NMR data

suggest that the ionic liquid can be easily removed from washing, which is expected given the highly porous nature of the hydrogel.

The final C-IPN gels continue to swell when immersed in water, equilibrating at a final weight that depends on the gel formulation. The formulation with highest solid content (C-IPN 1) swells to approximately two times its original weight, whereas the formulation with the lowest solid content (C-IPN 5) swells to nearly seven times its original weight (Supplementary Figure 8a). The gels can also be dried and re-swelled successfully in water. Interestingly, the final weight of the gels after re-swelling exceeds the equilibrium weight of as-synthesized C-IPN gels immersed in water, with the re-swelled weight ratio also dependent on the gel formulation. The formulation with highest solid content (C-IPN 1) has a final re-swollen weight that is 1.3 times its equilibrium swelled weight prior to drying, whereas the formulation with lowest solid content (C-IPN 5) has a final re-swollen weight that is nearly 6 times the equilibrium swelled weight (Supplementary Figure 8a). This observation suggests that drying the C-IPN hydrogels results in a reconfiguration of the polymer morphology that changes its swelling ability. Drying the hydrogels may encourage π–π stacking interactions between PEDOT:PSS that collapse the conducting polymer network into aggregates, leaving larger continuous void fractions for water to penetrate during the re-swelling process. This hypothesis is supported by the fact that PEDOT:PSS-only hydrogels made directly from mixing with ionic liquid also re-swell to $1.8 \pm 0.5$ times their original weight after drying. Importantly, the re-swelled gels retain the same high stretchability and low modulus of the as-made gels (Supplementary Figure 8b).

To characterize the mechanical properties of C-IPN hydrogels, stress strain measurements were recorded under tensile elongation. Based on the formulation, the C-IPN hydrogels have a wide range of elastic moduli ranging from 8 kPa to nearly 400 kPa (Table 1). Despite this wide range of stiffness, all C-IPN formulations could be stretched to over 100% strain before breaking (Fig. 3a, b). One reason for the excellent stretchability may be the high water content, which provides many degrees of freedom for polymer chains in the C-IPN to reorient in response to mechanical strain. In addition, the presence of hydrogen bonding from the PAAc network may provide sacrificial, dynamic bonds that can break to dissipate the stress from mechanical deformation, obviating the need to dissipate energy from fracture[33,34].

**Table 1 Different formulations of C-IPN and their corresponding electronic and mechanical properties**

| C-IPN formulation | AAc wt% | Bis/AAc wt ratio | Modulus (kPa) | Strain at break (%) | Conductivity (S m$^{-1}$) |
|---|---|---|---|---|---|
| 1 | 50 | 0.02 | 374 | 121 | 23.1 ± 5.6 |
| 2 | 33 | 0.01 | 175 | 163 | 23.7 ± 4.5 |
| 3 | 20 | 0.01 | 99 | 191 | 20.4 ± 4.7 |
| 4 | 20 | 0.002 | 23 | 399 | 12.7 ± 4.5 |
| 5 | 11 | 0.002 | 8 | 338 | 13.5 ± 2.3 |

Note that even though the modulus varies over three orders of magnitude, the conductivity in all cases stays above 10 S m$^{-1}$.

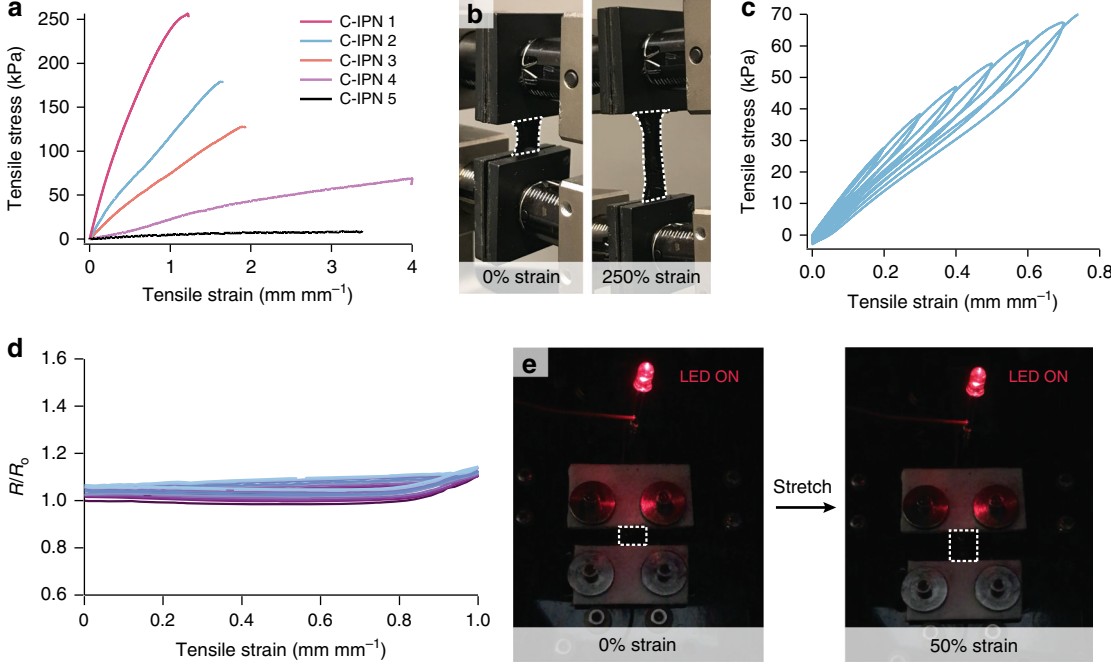

**Fig. 3** Mechanical and strain-dependent properties of C-IPN hydrogels. **a** Tensile elongation curves of different C-IPN formulations, showing that all formulations can be stretched to over 100% despite large differences in the elastic moduli, which is given by the initial slope of the stress/strain curve. **b** Picture of a C-IPN 4 gel being stretched to 250%. **c** Cyclic stress/strain tensile data for C-IPN 2 exhibiting minimal hysteresis, reflecting its excellent elastic properties. **d** Change in resistance, expressed as a ratio between resistance ($R$) and initial resistance ($R_o$), across a C-IPN 2 gel as it is cycled reversibly between 0 and 100% strain for 10 cycles. Despite the large changes in tensile strain, the resistance stays fairly constant near its initial value. **e** Due to the largely strain independent conductivity of the gels, it is able to keep an LED lit even after being stretched to 50% strain

Highly elastic conductive gels could also be fabricated using this method (Fig. 3c). This is particularly noteworthy, as elasticity has been a challenging property to incorporate into stretchable conducting polymeric materials. For instance, plasticizers have been used in literature to enhance the stretchability of dry PEDOT:PSS, but these materials still yield at relatively low strains < 20%, potentially limiting their usability as stretchable conductors in high-strain applications[29]. For instance, compared with the highly stretchable PEDOT:PSS/ionic liquid conductor developed in our group (Supplementary Figure 4), C-IPN gels exhibit considerably less hysteresis, while cycling the strain at the same conditions. The ability to orthogonally control mechanical properties using the PAAc network therefore enables us to take advantage of the elasticity of certain PAAc hydrogel formulations to impart elasticity onto the conductive composite.

Electrical conductivity of C-IPN gels was calculated from measuring the resistance of the gels using a four-point probe method. For all formulations, the conductivity was greater than $10 \, \text{S m}^{-1}$. The highest conductivity was $23 \pm 5.6 \, \text{S m}^{-1}$ and conductivity tended to decrease as a function of acrylic acid concentration in the PAAc hydrogel precursor solution (Table 1). The fact that the electrical conductivity could be maintained above $10 \, \text{S m}^{-1}$ despite large variations in elastic moduli is likely due to the ability for aqueous PEDOT:PSS solutions to form connected CP networks at a low weight concentration relative to acrylic acid. Indeed, the weight percent of PEDOT:PSS in the final gel is significantly lower than previously reported stretchable PEDOT:PSS hydrogels (Supplementary Table 1). Although connectivity of conducting domains is critical to achieving high electrical conduction[35], mechanical properties of composites, such as elastic modulus, are largely dictated by the relative amount of each component[36,37]. Thus, a composite material that is predominantly made up of a soft material will be soft, even if it contains a small amount of rigid domains. Analogously, because of the dilute microgel structure of PEDOT:PSS in aqueous solutions, the relatively small amount of PEDOT:PSS needed to form a connected conductive pathway in C-IPN enables its mechanical properties to be nearly orthogonally controlled by the non-conductive PAAc network. This represents a clear advantage over other conductive, stretchable hydrogels that have been synthesized by polymerizing EDOT within an electrically inert hydrogel matrix, wherein decent conductivities are only achieved with concentrations as high as 30 wt%.

Although the conductivity of C-IPN stays within the same order of magnitude despite significantly different elastic moduli, slight differences in conductivities can be attributed to differences in acrylic acid concentration, which is consistent with the different degrees of shrinking visually observed during the PAAc infiltration step. The fact that the conductivity does not vary significantly suggests that these differences in PEDOT:PSS gel strength are small compared to enhancement in conductivity that comes from forming a connected conductive network. This finding makes sense given that, in conductive composites, the most dramatic increase in conductivity typically occurs at the percolation threshold[38,39].

Given that the acidic monomers apparently interact with PEDOT:PSS, our infiltration method is required for them to effectively penetrate the PEDOT:PSS gel form a homogeneous blend. By contrast, directly blending PEDOT:PSS with acrylic acid resulted in rapid phase separation, probably due to the acidity of the monomers (Supplementary Figure 6a). Although the mixture could be polymerized to form a soft gel with moderate stretchability (Supplementary Figure 6b), the conductivity of the blend was significantly lower, at $0.21 \, \text{S m}^{-1}$. As expected, this value is in the same range of conductivities reported in the recent work of Wu et al.[15], who directly blended PEDOT:PSS with the

monomers used to make mechanically reinforcing inert secondary networks ($0.2–2.2 \, \text{S m}^{-1}$)[15]. In addition, the modulus and strain at break of PEDOT:PSS/PAAc hydrogels made by blending were lower than for hydrogels made with PAAc only, probably because a significant amount of monomer is lost when it crashes out with PEDOT:PSS upon blending (Supplementary Figure 6c). Thus, we believe that our infiltration method enables a greater degree of orthogonal control over the selection of the two interpenetrating networks in the conductive gel. This feature also enables us to incorporate alternative secondary networks, besides PAAc, into C-IPN. For instance, composite hydrogels made by infiltrating PEDOT:PSS gels with polyacrylamide (PAAm) precursors exhibited mechanical properties that closely resembled gels made with PAAm alone (Supplementary Figure 7).

Next, we characterized the change in electrical behavior under mechanical deformation by measuring the resistance across C-IPN while cycling back and forth between 0 and 50% strain (Fig. 3d). The electrical properties of the gel were surprisingly robust within this region, with the resistance staying approximately at its starting value throughout all 10 cycles. To verify that a high contact resistance did not mask the resistance change from the two-point test, four-point resistance measurements were also performed on the C-IPN after being stretched manually to multiple strains. Despite an apparent contact resistance contribution of $17 \, \Omega$ from the two-point measurement, the four-point stretching measurement confirmed that the resistance of the gels stayed largely unchanged even at large strains over 60% (Supplementary Figure 5). The robust conductivity may be attributed to both the connected nature of the PEDOT:PSS gel, as well as the ability for C-IPN to respond to deformation by re-orienting within the high water content environment or breaking hydrogen bonds in the PAAc network. These alternate dissipation mechanisms may enable the entangled colloidal PEDOT:PSS mesh to stay intact rather than disrupt percolation.

For bio-interfacing applications, the presence of ions in physiological solution can additionally contribute to the conductivity of C-IPN electrodes, particularly given the high liquid content of the hydrogels. To decouple ionic and electronic conductivities, we saturated C-IPN gels in $1 \times$ phosphate-buffered saline (PBS) solution and characterized their impedance using electrochemical impedance spectroscopy (EIS). The data were fit to the equivalent circuit model depicted in Fig. 4a, where $R_c$ represents ohmic resistance of the cell assembly, $R_e$ represents electronic resistance, $R_i$ represents ionic resistance, and $CPE_{dl}$ and $CPE_g$ are constant phase elements corresponding to the double-layer capacitance arising from ionic conduction and the geometric capacitance of the gel, respectively[40–43]. The distorted semicircular shape of the Nyquist plot suggests the presence of comparable ionic and electronic conductivities (Fig. 4b, d)[44]. The parameters extracted from fitting the data to the equivalent circuit model confirms that the ionic and electronic resistances are within the same order of magnitude (Fig. 4e). To validate this model, we measured the DC voltage as a function of time for a constant applied current of 5 mA on the C-IPN 2 sample (Fig. 4c)[45]. The plateau value of the voltage, 0.0247 V, corresponds to the electronic contribution to the resistance only, and the $R_e$ value extrapolated by this method ($4.94 \, \Omega$) is comparable to the value from fitting the AC data ($4.97 \, \Omega$) to within < 1% error. Finally, we characterized the impact of different C-IPN formulations on mixed conductivity by performing the EIS measurement on three formulations of varying density, with C-IPN 1 being the most dense and C-IPN 3 being the least dense (Fig. 4d). As the density of the gels decreases, the relative contribution of ionic resistance to the overall impedance decreases as well (Fig. 4e). This is reasonable given that the hydrogel's density is directly proportional to the electronic conductivity (Table 1), whereas it is

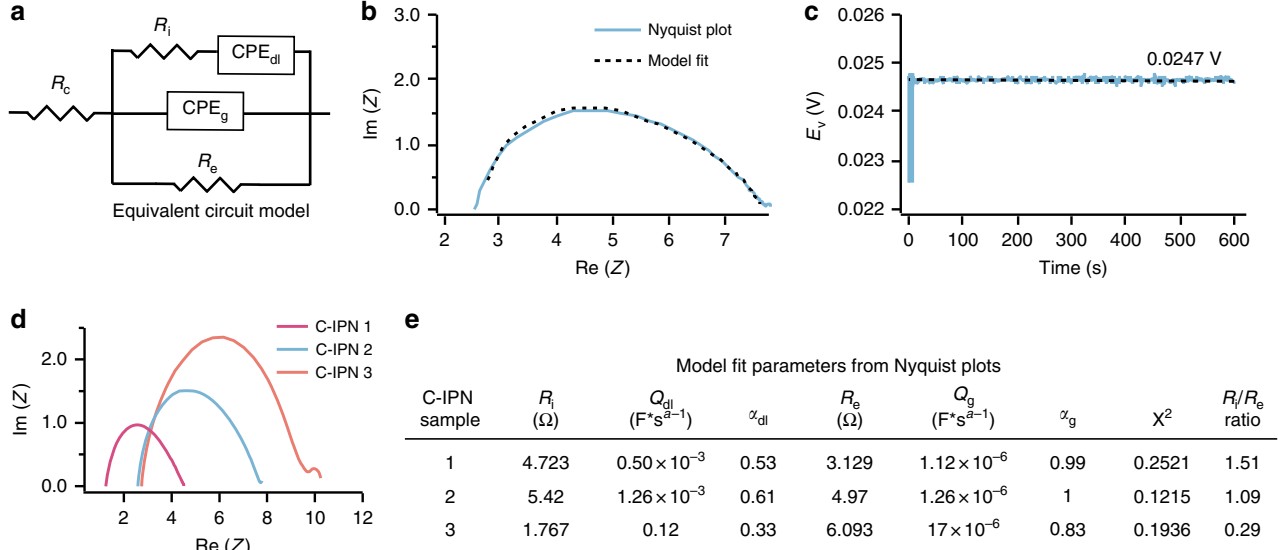

| C-IPN sample | $R_i$ (Ω) | $Q_{dl}$ (F*s$^{a-1}$) | $\alpha_{dl}$ | $R_e$ (Ω) | $Q_g$ (F*s$^{a-1}$) | $\alpha_g$ | $X^2$ | $R_i/R_e$ ratio |
|---|---|---|---|---|---|---|---|---|
| 1 | 4.723 | $0.50 \times 10^{-3}$ | 0.53 | 3.129 | $1.12 \times 10^{-6}$ | 0.99 | 0.2521 | 1.51 |
| 2 | 5.42 | $1.26 \times 10^{-3}$ | 0.61 | 4.97 | $1.26 \times 10^{-6}$ | 1 | 0.1215 | 1.09 |
| 3 | 1.767 | 0.12 | 0.33 | 6.093 | $17 \times 10^{-6}$ | 0.83 | 0.1936 | 0.29 |

**Fig. 4** Characterization of dual ionic and electronic conductivity. **a** Equivalent circuit model representing the bulk C-IPN hydrogel. $R_e$ represents electronic resistance, $R_i$ represents ionic resistance, $CPE_{dl}$ represents the double-layer capacitive phase element (CPE), whereas $CPE_g$ represents the geometric CPE. CPE elements are used to account for inhomogeneous or imperfect capacitance, and are represented by the parameters $Q$ and $\alpha$, where $Q$ is a pseudocapacitance value and $\alpha$ represents its deviation from ideal capacitive behavior. The true capacitance ($C$) can be calculated from these parameters by the relationship $C = Q\,\omega_{max}{}^{\alpha-1}$, where $\omega_{max}$ represents the frequency at which the imaginary component reaches a maximum[43]. $R_c$ represents the total ohmic resistance of the cell assembly. **b** Nyquist plot obtained from performing electrochemical impedance spectroscopy (EIS) through a bulk C-IPN 2 gel, overlaid with the plot predicted from the equivalent circuit model. Impedance was measured between 500 mHz and 7 MHz, with higher real components of the impedance obtained at lower frequencies. **c** When a constant DC current of 5 mA is applied through the C-IPN 2 gel, the voltage across the gel plateaus to a value of 0.0247 V. This value can be used to calculate an electronic resistance that is comparable to the value of $R_e$ extracted from the model. **d** Overlay of Nyquist plots obtained for three C-IPN formulations, where C-IPN 1 is the stiffest and densest, and C-IPN 3 is the softest and least dense. Impedance was measured between 500 mHz and 7 MHz, with higher real components of the impedance obtained at lower frequencies. **e** Values for all relevant parameters extracted for the three C-IPN formulations by fitting their EIS data with the equivalent circuit model. As the gel stiffness and density increase, the relative ionic resistance within the gel increases as well

inversely proportional to the total volume of electrolyte, which facilitates ion transport, in the hydrogel.

## Discussion

We have presented a method for fabricating highly conductive hydrogels with dual electronic and ionic conductivity and highly tunable mechanical properties that mimic biological tissue. This combination of properties makes C-IPN hydrogels promising for integration into wearable and implantable devices, for which the ability to couple high electronic conductivity with low modulus and high stretchability is particularly desired at soft biological interfaces such as the brain. In addition, the ability to tune our gel's mechanical properties without compromising its conductivity makes it an attractive materials platform for tissue engineering and cell culture, since it is well known that cells are highly responsive to the mechanical properties of their surrounding environment[46,47]. C-IPN hydrogels offer a route to enable electrical stimulation and recording while preserving the appropriate three-dimensional architecture and matched mechanical properties needed to mimic human tissue in vitro and to support cell viability.

In the future, C-IPN can be integrated into more application-specific devices by taking advantage of the fact that it can be easily molded into different shapes and geometries for different target applications. Although there may be a lower resolution limit given by the approximate PEDOT:PSS microgel size of 250 nm[28], sub-micrometer patterning is unnecessary for many biomedical applications. For instance, deep-brain stimulation electrodes typically have surface areas on the order of 1 mm$^2$ [48]. By contrast, we have demonstrated size resolution down to 10 μm using a

simple mold casting method. Furthermore, the fact that C-IPN can be dried and re-swelled means that it can be compatibly integrated with other materials that may require dry or non-aqueous environments for processing. With their processability, mechanical tunability, and excellent electronic properties, C-IPN gels are a highly versatile electronic material for future bio-interfacing applications.

## Methods

**Materials**. All chemicals were purchased from Sigma-Aldrich. PEDOT:PSS Orgacon ICP 1050 was provided by Agfa as a surfactant-free aqueous dispersion with 1.1 wt% solid content. The PEDOT:PSS dispersion was filtered through a 1.0 μm Nylon filter to remove any large agglomerates prior to use. The ionic liquid 4-(3-Butyl-1-imidazolio)-1-butanesulfonic acid triflate (19597 Aldrich, CAS 439937-63-0) was used to induce gelation of PEDOT:PSS. The thermal initiator used to initiate radical polymerization of the PAAc precursors was 2,2'-Azobis(2-methyl-propionamidine) dihydrochloride (440914 Aldrich, CAS 2997-92-4), a water-soluble azo-initiator that can be initiated above 60 °C. As-purchased acrylic acid (147230 Aldrich, CAS 79-10-7) was run through a basic alumina plug to remove MEHQ inhibitor. N,N'-methylenebis(acrylamide) (146072 Sigma-Aldrich, CAS 110-26-9) was used as the crosslinking monomer.

**Synthesis of C-IPN hydrogels**. To induce gelation of PEDOT:PSS, ionic liquid was slowly added to the filtered PEDOT:PSS dispersion while stirring. Subsequently, the PEDOT:PSS/ionic liquid mixtures were poured into molds, briefly degassed, and then sealed and placed in a 70 °C oven for 12 h to ensure complete gelation. Next, the PEDOT:PSS hydrogels were immersed in DI water, which was exchanged a total of three times over the course of 24 h. Then, the washed PEDOT:PSS hydrogels were immersed in the PAAc precursor solution, which was also exchanged a total of three times over the course of 24 h. Finally, the soaked PEDOT:PSS hydrogels were sealed and placed in a 70 °C oven to polymerize the PAAc network for 30 min to form C-IPN. After this step, the C-IPN was thoroughly rinsed in water to remove any unreacted acrylic acid.

**Microscopy**. SEM images of C-IPN hydrogels were obtained lyophilizing the hydrogels. Images were obtained using an FEI XL30 Sirion SEM.

**Characterization of mechanical properties**. Rheological characterization of PEDOT:PSS hydrogels was performed using a TA Instruments ARES-G2 rheometer. For frequency sweeps, a 25 mm parallel plate geometry was used with a 1 mm gap size at a constant temperature of 37 °C and strain of 1%. For the gelation time measurement, a 40 mm parallel plate was used instead to improve the signal quality, as the samples were liquid-like. Measurements were taken at 25 °C at 1 Hz and a strain rate of 1%. Tensile elongation measurements of PAAc and C-IPN hydrogels were performed using an Instron 5565 at a strain rate of 10% per minute. For cycling tests, samples were strained at a rate of 50% per minute and allowed to equilibrate for 5 min between each stretching cycle.

**Characterization of electronic properties**. Conductivity was calculated by measuring resistance using a standard four-point probe head. Reported values reflect an average over a minimum of three to five measurements obtained for each condition. To continuously measure the resistance while stretching, samples were attached to a homemade stretching station attached to a Keithley multimeter by electrical leads at both ends. Samples were cycled to various strains at a constant strain rate of 10%/min. To measure the resistance change using a 4-point probe method, samples were manually stretched, with initial and final lengths noted using a ruler. At each strain, the resistance of the sample was measured using a standard four-point probe head.

EIS measurements were conducted using a Bio-Logic VSP potentiostat. A punching tool was used to make gel samples with a constant cross-sectional area. The samples were sandwiched between two platinum electrodes within a Swagelok cell and surrounded by 1 × PBS, to ensure samples had a consistent degree of electrolyte saturation. AC impedance measurements were obtained between 500 mHz and 7 MHz at an open-circuit potential of 20 mV amplitude. The impedance data were fit using the Zfit tool from Bio-Logic's EC-Lab software. For DC measurements, a constant current of 5 mA was applied through the sample.

**Chemical characterization**. NMR spectra were recorded on a Varian mercury console spectrometer at 400 MHz at room temperature. Chemical shifts for $^1$H-NMR spectra were reported in parts per million and were referenced to residual protonated solvent $(CD_3)_2SO$: $\delta$ 2.50). Infrared spectroscopy was performed on a Nicolet iS50 FTIR Spectrometer with a diamond ATR crystal and DTGS detector. XPS was performed using a PHI Versaprobe III Scanning XPS Microprobe.

**Data availability**. All relevant data is available upon request from the authors.

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

## Acknowledgements

This work was supported by the Precourt Institute for Energy (PIE) grant at Stanford University. Part of this work was performed at the Stanford Nano Shared Facilities (SNSF), supported by the National Science Foundation under award ECCS-1542152. V.R.F. was supported by the Department of Defense (DoD) through the National Defense Science & Engineering Graduate (NDSEG) Fellowship Program. H.T. was supported by an appointment to the Intelligence Community Postdoctoral Research Fellowship Program at Stanford University, administered by Oak Ridge Institute for Science and Education through an interagency agreement between the U.S. Department of Energy and the Office of the Director of National Intelligence. M.L. was partially supported by the Postdoctoral Fellowship from the National Research Foundation of Korea under Grant Number NRF-2017R1A6A3A03007053.

## Author contributions

V.R.F., H.T., and Z.B. developed the concept. V.R.F. and H.T. synthesized and characterized the materials. M.L. developed the electrochemical impedance spectroscopy method. V.R.F. wrote the manuscript with contributions from all co-authors.

## Additional information

**Competing interests:** The authors declare no competing interests.

