## [Peer Review File · Nature Communications]

Reviewers' comments:

Reviewer #1 (Remarks to the Author):

The work presented by Feig et al reports about a new PEDOT:PSS based hydrogel, with high electronic conductivity and orthogonally tuneable elastic properties, that could have great impact on the biomedical field, due to the matching of mechanical modulus with biological tissue. The works clearly present a relevant advancement of the state of the art of the sector, while being interesting for a broad audience of readers. State of the art is clearly presented and well addressed. The manuscript is well presented and easy to follow. Technical and scientific quality of the presented results seems to be adequate for the targeted journal.

In my opinion, just a few minor issues should be addressed by the authors before be considered for publication, as follow:

- Figure 3C. I guess there is a mistake with measurement unit; I think that the right one should be kPa, instead of MPa.
- Figure 3D. Tensile strain is here reported in %, while in the rest of paper is reported in mm/mm. Please uniform the notation among all the paper and supplementary info.
- Figure 4B and 4D. Graphs could be more informative if starting and ending frequency are specified directly on the graph (or at least in the caption).
- Figure 4E. Some of the model fit parameters reported are not specified in the main text, neither on the caption. Please add description for all the parameters.
- Figure S2B. Please add scale bar.
- Please use uniform notation for Tensile Stress among all the figures (including supplementary ones). (es. MPa or kPa) Moreover in the axis, please avoid to use 6×10^{-2} ... instead of 0.02).
- Figure S5 and S8. Please add error bars for standard deviation.

Reviewer #2 (Remarks to the Author):

The manuscript of Feig and co workers presents an impressive advance in conducting hydrogel materials. The authors show low PEDOT:PSS loading by using ionic liquids to promote microgel particle attraction, forming percolating hydrogel networks. They then infiltrate with PAAc, initiator and cross linker to tune the mechanical properties of the gel. The impressive findings show that the conductivity (~ 10 - 20 S/m) is maintained with 100-200% strain. I believe that while the findings are of interest to those working on conducting hydrogels, the work may lack the generality or demonstration of application necessary to appeal to the broad audience of Nature Communications.

The paper is very clearly written, and experiments are performed carefully and analyzed thoughtfully. Especially the EIS analysis of mixed conduction. However, the work feels in line with others published in more specialized materials journals. A logical extension would be to show that this approach could work with other conducting polymer dispersions, or with other interpenetrating polymer types.

While the authors discuss applications in neurostimulation (especially with materials that can maintain performance and take the required strains of brain motions) they do not show potential results in this realm, nor biocompatibility.

Reviewer #3 (Remarks to the Author):

Feig et al. described a very interesting material where wet hydrogel network, high conductivity, high stretchability and low elastic moduli were nicely integrated. The team demonstrated a series of unique features in these conductive hydrogel materials. For example, they explored the dynamic

hydrogen bonds that promote the high stretchability while still maintaining the unprecedented conductivity in PEDOT/PSS-based hydrogels. This referee suggests acceptance of this manuscript after the authors address the following minor concerns:

1. It would be better to show a high resolution SEM for Fig. 1E. The current one presents limited information to readers.
2. The authors should add the scale bars in Fig. 1B.
3. What is the reason that caused the significant changes in the loss moduli near 100 rad/s for all curves in Fig. 2B? Please clarify this in the manuscript.
4. The authors missed the figure captions for Figs. 4D and 4E.
5. Please include error bars in Figs. S5 and S8A.

Reviewer #1:

The work presented by Feig et al reports about a new PEDOT:PSS based hydrogel, with high electronic conductivity and orthogonally tuneable elastic properties, that could have great impact on the biomedical field, due to the matching of mechanical modulus with biological tissue. The works clearly present a relevant advancement of the state of the art of the sector, while being interesting for a broad audience of readers. State of the art is clearly presented and well addressed. The manuscript is well presented and easy to follow. Technical and scientific quality of the presented results seems to be adequate for the targeted journal.

We would like to thank the reviewer for these favorable comments on the manuscript, and for kindly providing detailed suggestions to improve the manuscript quality. Below, please find our responses to each individual point raised by the reviewer:

- **Figure 3C. I guess there is a mistake with measurement unit; I think that the right one should be kPa, instead of MPa.**

That is correct; we have changed the measurement unit on the y-axis to kPa.

- **Figure 3D. Tensile strain is here reported in %, while in the rest of paper is reported in mm/mm. Please uniform the notation among all the paper and supplementary info.**

We have changed all the tensile strain data in the manuscript to be reported in units of mm/mm. In addition to Figure 3D, that also includes Figure S5 and Figure S8B.

- **Figure 4B and 4D. Graphs could be more informative if starting and ending frequency are specified directly on the graph (or at least in the caption).**

The following sentence was added to the caption describing Figure 4B and Figure 4D: *“Impedance was measured between 500 mHz and 7 MHz, with higher real components of the impedance obtained at lower frequencies.”*

- **Figure 4E. Some of the model fit parameters reported are not specified in the main text, neither on the caption. Please add description for all the parameters.**

We used the Zfit tool from the software EC-Lab to fit the data to the equivalent circuit model shown in Figure 4A. Accordingly, we have added the following sentence to the Methods section: *“The impedance data were fit using the Zfit tool from Bio-Logic’s EC-Lab software.”*

- **Figure S2B. Please add scale bar.**

We have added scale bars to both the images in Figure S2B.

- **Please use uniform notation for Tensile Stress among all the figures (including supplementary ones). (es. MPa or kPa) Moreover in the axis, please avoid to use 6×10^{-2} ... instead of 0.02).**

We have changed all Tensile Stress data to be represented in kPa. Specifically, the following graphs were changed: Figure 3A, Figure S1, Figure S7, and Figure S8B.

- Figure S5 and S8. Please add error bars for standard deviation.

In both of these figures, each data point corresponds to a single measurement that corroborates data and observations in the main text. Figure S5 corroborates the data in Figure 3D, which was repeated for 10 cycles. Figure S8A corroborates the qualitative observation that our materials can swell and re-swell, and that the swelling behavior is dependent on the gel formulation.

Reviewer #2:

The manuscript of Feig and co workers presents an impressive advance in conducting hydrogel materials. The authors show low PEDOT:PSS loading by using ionic liquids to promote microgel particle attraction, forming percolating hydrogel networks. They then infiltrate with PAAc, initiator and cross linker to tune the mechanical properties of the gel. The impressive findings show that the conductivity (~10-20 S/m) is maintained with 100-200% strain. I believe that while the findings are of interest to those working on conducting hydrogels, the work may lack the generality or demonstration of application necessary to appeal to the broad audience of Nature Communications. The paper is very clearly written, and experiments are performed carefully and analyzed thoughtfully. Especially the EIS analysis of mixed conduction. However, the work feels in line with others published in more specialized materials journals. A logical extension would be to show that this approach could work with other conducting polymer dispersions, or with other interpenetrating polymer types. While the authors discuss applications in neurostimulation (especially with materials that can maintain performance and take the required strains of brain motions) they do not show potential results in this realm, nor biocompatibility.

We would like to thank the reviewer for these favorable comments on the manuscript. Below, we would like to respond to the two concerns raised by the reviewer.

1. “I believe that while the findings are of interest to those working on conducting hydrogels, the work may lack the generality or demonstration of application necessary to appeal to the broad audience of Nature Communications.”

and

“However, the work feels in line with others published in more specialized materials journals.”

Thank you for pointing out this concern. We believe that our hydrogel is of broad interest across numerous diverse applications, and that submitting to a more specialized journal may exclude access by certain researchers who could benefit from the work. Particularly, our work may be of interest to researchers working on wearable and implantable electronic devices, as well as researchers working with tissue engineering and cell culture. Beyond that, researchers in other fields may be interested in some of the unique secondary features highlighted in our manuscript, beyond mechanical properties; for instance, researchers developing organic electrochemical transistors (OECTs) may be interested in our characterization of our hydrogel's dual ionic and electronic conductivity. These varied research communities are quite disparate, and it would be difficult to reach the appropriate broad audience in a more specialized journal. In light of this, we believe that *Nature Communications* is an ideal platform from which to showcase our work, especially because it is an open access publication.

We agree with the reviewer that there is an opportunity to clarify the generality of our work, and we have reworked the Discussion section accordingly:

3. Discussion

We have presented a novel method for fabricating highly conductive hydrogels with dual electronic and ionic conductivity and highly tunable mechanical properties that mimic biological tissue. This combination of properties makes C-IPN gels promising for integration into wearable and implantable devices, for which the ability to couple high electronic conductivity with low modulus and high stretchability is particularly desired at soft biological interfaces like the brain. Additionally, the ability to tune mechanical properties without compromising conductivity makes C-IPN an attractive materials platform for tissue engineering and cell culture, since it is well known that cells are highly responsive to the mechanical properties of their surrounding environment.^{41,42} C-IPN gels offer a novel route to enable electrical stimulation and recording while preserving the appropriate 3D architecture and matched mechanical properties needed to mimic human tissue *in vitro* and to support cell viability.

In the future, C-IPN gels can be integrated into more application-specific devices by taking advantage of the fact that they can be easily molded into different shapes and geometries for different target applications. While there may be a lower resolution limit given by the approximate PEDOT:PSS microgel size of 250 nm,²⁶ sub-micron patterning is unnecessary for many biomedical applications. For instance, deep brain stimulation electrodes typically have surface areas on the order of 1 mm².⁴³ By contrast, we have demonstrated size resolution down to 10 μm using a simple mold casting method. Furthermore, the fact that C-IPN can be dried and re-swelled means that it can be compatibly integrated with other materials that may require dry or non-aqueous environments for processing. With their processability, mechanical tunability, and excellent electronic properties, C-IPN gels are a highly versatile electronic material for future bio-interfacing applications.

We thank the reviewer again for raising this concern and giving us the chance to greatly enhance the quality of the manuscript.

2. “While the authors discuss applications in neurostimulation (especially with materials that can maintain performance and take the required strains of brain motions) they do not show potential results in this realm, nor biocompatibility.”

The biocompatibility of PEDOT:PSS and polyacrylic acid have already been well-established in literature.¹⁻⁵ We have further emphasized this in the main text, on page 1 in the Introduction and page 3:

- “Among these CPs, PEDOT:PSS is advantageous because of its biocompatibility” (page 1)
- “...we selected polyacrylic acid (PAAc) as a secondary polymer network due to its biocompatibility^{28,29} and high concentration of hydrogen bonding, which we hypothesized could further mechanically reinforce the C-IPN” (page 3)

We chose not to specifically demonstrate results relating to neurostimulation because we believe that our material and fabrication method can be generally useful to a broader audience. Furthermore, we are in the process of conducting a more extensive study on the use of our materials for biological stimulation and recording, and hope to report the findings in a more conclusive follow-up report.

We thank the reviewer again for sharing their thoughtful feedback, and we hope we have addressed their concerns to their satisfaction.

1. H. L. Seldon et al, “Silastic with polyacrylic acid filler: swelling properties, biocompatibility and potential use in cochlear implants.” *Biomaterials* (1994), 15, 14: 1161-1169.
2. Y.-H. Ma et al, “Magnetically targeted thrombolysis with recombinant tissue plasminogen activator bound to polyacrylic acid-coated nanoparticles.” *Biomaterials* (2009), 30, 19: 3343-3351.
3. M. Yazdimaghani et al, “Biomineralization and biocompatibility studies of bone conductive scaffolds containing poly(3,4-ethylenedioxythiophene):poly(4-styrene sulfonate) (PEDOT:PSS).” *J. Mater. Sci.: Materials in Medicine* (2015), 26: 274.

4. M. Berggren and A. Richter-Dahlfors, "Organic Bioelectronics." *Adv. Mater.* (2007), 19: 3201-3213.
5. E. De Giglio et al, "Analytical investigations of poly(acrylic acid) coatings electrodeposited on titanium-based implants: a versatile approach to biocompatibility enhancement." *Analytical and Bioanalytical Chemistry.* (2007), 389, 7-8: 2055-2063

Reviewer #3:

Feig et al. described a very interesting material where wet hydrogel network, high conductivity, high stretchability and low elastic moduli were nicely integrated. The team demonstrated a series of unique features in these conductive hydrogel materials. For example, they explored the dynamic hydrogen bonds that promote the high stretchability while still maintaining the unprecedented conductivity in PEDOT/PSS-based hydrogels.

We would like to thank the reviewer for these favorable comments on the manuscript, and for kindly providing detailed suggestions to improve the manuscript quality. Below, please find our responses to each individual point raised by the reviewer:

1. It would be better to show a high resolution SEM for Fig. 1E. The current one presents limited information to readers.

We replaced Fig. 1E with a higher resolution SEM image, and edited the caption to emphasize that the SEM highlights the porosity and uniform cross-section of the material.

2. The authors should add the scale bars in Fig. 1B.

We have added scale bars to each of the images in Fig. 1B.

3. What is the reason that caused the significant changes in the loss moduli near 100 rad/s for all curves in Fig. 2B? Please clarify this in the manuscript.

The shape of the loss modulus is characteristic of the frequency dependent behavior within the rubbery regime of a typical viscoelastic material, since the region is bounded by two maxima in the loss modulus curve. The kinks at 100 rad/s may represent an artifact from the measurement setup. Regardless, we believe that the most important takeaways from this figure arise from the shape and concentration dependency of the storage modulus curves, which directly correspond to the crosslinking density of the gel, and which do not appear to have this kink at 100 rad/s.

4. The authors missed the figure captions for Figs. 4D and 4E.

We believe the reviewers are referring to Figs 3D and 3E, and we apologize for neglecting to include the captions originally. The captions have now been included, and read as follows:

“(D) Change in resistance across a C-IPN 2 gel as it is cycled reversibly between 0% and 100% strain for 10 cycles. Despite the large changes in tensile strain, the resistance stays fairly constant near its initial value. (E) Due to the largely strain independent conductivity of the gels, it is able to keep an LED lit even after being stretched to 50% strain.”

5. Please include error bars in Figs. S5 and S8A

In both of these figures, each data point corresponds to a single measurement that corroborates data and observations in the main text. Figure S5 corroborates the data in Figure 3D, which was repeated for 10 cycles. Figure S8A corroborates the qualitative observation that our materials can swell and re-swell, and that the swelling behavior is dependent on the gel formulation.

REVIEWERS' COMMENTS:

Reviewer #1 (Remarks to the Author):

The authors correctly fulfilled all the doubts I pointed out in the previous turn of revision. In my opinion the paper is now suitable for publication.

Reviewer #2 (Remarks to the Author):

The authors have done a good job addressing many of the comments of the reviewers. Regarding one particular comment, I feel that the authors have not addressed the outstanding request to better describe or define the fit parameters used to model the data (for example, the authors do not define "Q", or "a" in text. What/where is the phase term?

On the general level, I must point out that I feel that authors do not address some relevant recent work, especially from the groups of Molly Stevens and Rylie Green that are relevance to the current manuscript. For example, one discusses interpenetrating conducting hydrogels, etc. While the method discussed in this work is slightly different, I note that these works usually also show some biological compatibility, or go a bit further beyond the synthesis and characterization, and at that are published in more chemistry and materials related journals:

<https://pubs.acs.org/doi/abs/10.1021/acs.chemmater.6b01298>

<https://onlinelibrary.wiley.com/doi/abs/10.1002/adhm.201601177> I do appreciate the novelty of some aspects of this work, and would recommend publication of this work in Nature Communications if the authors can do a more thorough job describing how this work fits in with recent publications (over the past 2-3 years) and differentiating/highlighting the significance with respect to these works.

Reviewer #3 (Remarks to the Author):

The authors have addressed the concerns from this referee. Acceptance is recommended.

Response to Reviewers' comments:

We would like to thank Reviewer #2 for continuing to provide thoughtful feedback to help us improve the quality of our manuscript. Please find below our point-by-point response detailing how we have addressed their comments in our paper.

The authors have done a good job addressing many of the comments of the reviewers. Regarding one particular comment, I feel that the authors have not addressed the outstanding request to better describe or define the fit parameters used to model the data (for example, the authors do not define “Q”, or “a” in text. What/where is the phase term?

We apologize to the reviewer for missing this in our previous response. Q and α are parameters used to describe the constant phase elements (CPEs) in our equivalent circuit model. We have added the following sentence to the caption of Figure 4A, where the impedance model fit data are presented, to clarify this:

“CPE elements are used to account for inhomogeneous or imperfect capacitance, and are represented by the parameters Q and α , where Q is a pseudocapacitance value and α represents its deviation from ideal capacitive behavior. The true capacitance (C) can be calculated from these parameters by the relationship $C = Q \omega_{max}^{\alpha-1}$, where ω_{max} represents the frequency at which the imaginary component reaches a maximum.”

On the general level, I must point out that I feel that authors do not address some relevant recent work, especially from the groups of Molly Stevens and Rylie Green that are relevance to the current manuscript. For example, one discusses interpenetrating conducting hydrogels, etc. While the method discussed in this work is slightly different, I note that these works usually also show some biological compatibility, or go a bit further beyond the synthesis and characterization, and at that are published in more chemistry and materials related journals:

<https://pubs.acs.org/doi/abs/10.1021/acs.chemmater.6b01298>

<https://onlinelibrary.wiley.com/doi/abs/10.1002/adhm.201601177>

We thank the reviewer for pointing out these relevant recent works and we have now included references to them in the Introduction section of the paper. Both of these works rely on the polymerization of PEDOT using an alternative dopant to PSS. By contrast, one of the major advantages of our work is that we can directly use commercially available PEDOT:PSS, wherein PEDOT is doped in a reproducibly and highly conductive form. This feature is particularly important to enable widespread usability of our conducting hydrogel to researchers in fields who may not have the facilities or experience to perform chemical synthesis. To emphasize this, we have referenced these two papers in the following modified sentence in our Introduction:

“Among these CPs, PEDOT:PSS is advantageous because of its biocompatibility, and because it is commercially available in its doped form with reproducibly high conductivity, thus avoiding problems with batch to batch variation from in situ polymerization²⁵ and the use of alternative dopants that are less effective than PSS.²⁶ Reported examples of stretchable conductive hydrogels based on PEDOT:PSS either rely on in situ polymerization of EDOT within an inert hydrogel matrix,^{13,14,27} or involve blending PEDOT:PSS with hydrogel-forming precursors.¹⁵ However, these strategies suffer from low conductivity (0.01 - 2.2 S m^{-1}) with relatively high PEDOT:PSS content up to 30 wt%, which makes it difficult to selectively tune the mechanical properties of the gel.¹³⁻¹⁵”

In addition, our high electrical conductivity is achieved by ensuring conducting polymer connectivity via the formation of a gel network, and the gelation mechanism we discuss is dependent on the unique solution behavior of PEDOT doped with PSS. The unique dilute microgel structure of aqueous PEDOT:PSS dispersions is also what enables us to achieve gels at very low solids concentrations, which is critical to enabling orthogonal control of mechanical properties using a secondary polymer network. We have added the following sentence to the Introduction to reemphasize this point:

“Still, the ability to form gels directly from commercially available PEDOT:PSS solutions presents a unique opportunity to control CP network connectivity to improve electronic conductivity.”

This point is also discussed in more detail in Section A of our Results section, where we reference the recent fundamental insights into PEDOT:PSS solution structure published from the group of Prof. Murugappan Muthukumar. We have also included the following modification in the Results section to emphasize this point:

“Analogously, because of the dilute microgel structure of PEDOT:PSS in aqueous solutions, the relatively small amount of PEDOT:PSS needed to form a connected conductive pathway in C-IPN enables its mechanical properties to be nearly-orthogonally controlled by the non-conductive PAAc network.”

We hope we have been able to address the reviewer’s concerns, and we would like to thank the reviewer again for taking the time to provide their helpful input throughout the entirety of this review process.

Regards,

Vivian R. Feig, Helen Tran, Minah Lee, and Zhenan Bao